# Epigenetic Biomarkers for Cervical Cancer Progression: A Scoping Review

**DOI:** 10.3390/ijms26199423

**Published:** 2025-09-26

**Authors:** Efthymios Ladoukakis, Gracia Andriamiadana, Fatema Hajizadah, Lewis G. E. James, Belinda Nedjai

**Affiliations:** 1Molecular Epidemiology Laboratory, Wolfson Institute of Population Health, Queen Mary University of London, London EC1M 6BQ, UK; m.ladoudakis@qmul.ac.uk; 2Centre for Advanced Cardiovascular Imaging, William Harvey Research Institute, NIHR Barts Biomedical Research Centre, Queen Mary University of London, London EC1M 6BQ, UK; g.andriamiadana@qmul.ac.uk; 3Centre for Neuroscience, Surgery and Trauma, Blizard Institute, Queen Mary University of London, London E1 2AT, UK; f.hajizadah@qmul.ac.uk; 4Centre for Cancer Biomarkers and Biotherapeutics, Barts Cancer Institute, Queen Mary University of London, London EC1M 6AU, UK; lewis.james@qmul.ac.uk

**Keywords:** epigenomics, cervical cancer, biomarkers, mtDNA, lncRNA, circRNA, piRNA, miRNA, histone modifications, repetitive elements, transposable elements

## Abstract

Cervical cancer remains the fourth most common cancer among women globally, disproportionately impacting low- and middle-income countries despite the existence of HPV vaccines. While DNA methylation has been studied extensively as a biomarker, other epigenetic mechanisms remain underexplored. This scoping review aims to report such underexplored epigenetic biomarkers linked to cervical cancer, shifting the focus beyond global nuclear DNA methylation. Literature searches were performed using Google Scholar via Publish or Perish software including studies published until January 2025. Our review focused on mitochondrial DNA, non-coding RNA, histone modifications, and repetitive elements. Mitochondrial DNA methylation has been proposed as a cervical cancer biomarker, although supporting evidence is limited. Histone modifications are more consistently reported to be involved both in cervical cancer onset and aggressiveness. Similarly, aberrant expression of lncRNAs, circRNAs, miRNAs, and piRNAs has been associated with poor prognosis. Finally, hypomethylation in repetitive elements such as LINE-1 and Alu is often observed in cervical cancer, contributing to genomic instability and tumorigenesis. Highlighting these alternative epigenetic mechanisms, our review emphasizes the importance of expanding biomarker discovery beyond the traditional nuclear DNA methylation. Understanding these mechanisms may improve early detection and personalized disease management strategies for cervical cancer.

## 1. Introduction

Cervical cancer is the fourth most common and lethal female malignancy worldwide, disproportionately affecting women in low and middle-income countries (LMICs) [1]. The vast majority of cervical cancers are caused by “high-risk” subtypes of human papillomavirus (HPV), particularly types 16 and 18 [2]. As HPV is a sexually transmitted virus, HPV vaccination, safe sexual practices, and other methods of avoidance of HPV infection prevent the spread of the disease, contributing to reduced incidence and mortality rates of cervical cancer [2]. Additionally, secondary preventative methods, such as Pap smear and HPV screening tests, allow close monitoring of the cervical tissue condition, with the former identifying cervical intraepithelial neoplasias (CIN) at early stages and the latter being an indirect biomarker of disease risk [2,3]. Diagnosing pre-cancerous and cancerous epithelial changes at an early stage could increase the likelihood of effective treatment and curing of the disease. However, for this to be possible, a deeper understanding of the disease’s underlying pathological and etiological mechanisms is required. For instance, while HPV infection is the common denominator in most cervical cancer cases, the majority of HPV infections do not lead to cervical cancer, which means that there are additional molecular factors that can contribute to the initiation and progression of the disease [2]. Such factors include epigenetic mechanisms which play a key role in the development of many types of cancer [4], including cervical [5,6], as well as in the host immune response. For example, epigenetic dysregulation of immune checkpoint molecules Tim-3 and galectin-9 can contribute to immune evasion during cervical cancer [7]. Nuclear DNA methylation (nDNA) is evidently one of the most well-characterized epigenetic mechanisms and is proven in many instances to be associated with CIN onset and its progression to invasive cervical cancer. In fact, both host [8] and viral genome methylation patterns [9] have been highly associated with disease occurrence and progression.

The immense potential of DNA methylation biomarkers to predict disease progression at early stages has been demonstrated in numerous studies [5,6,10,11,12,13]. In the clinic, application of DNA methylation-based assays has shown similar or greater sensitivity and specificity to detect CIN2+ and CIN3+ than standard cytology and HPV16/18 genotyping [14]. Exploring alternative epigenetic mechanisms as potential biomarkers may provide a complementary tool that can be used either standalone or in combination with nuclear DNA methylation for multi-omic triage tests. The epigenetic landscape in relation to cancer has been studied increasingly in recent years [15] and as a result, additional epigenetic mechanisms, besides nuclear DNA methylation, have been identified as key factors that can drive disease outcomes. Histone modifications, non-coding RNAs (ncRNAs), and even mitochondrial DNA (mtDNA) have been associated with many types of cancer, including cervical. Recent advances in the high throughput sequencing technologies, as well as emerging methodologies such as Cleavage Under Targets and Tagmentation (CUT&Tag) and Assay for Transposase-Accessible Chromatin using sequencing (ATAC-seq), have allowed us to access new epigenetic layers, thus uncovering novel candidates with biomarker potential. However, these potential biomarkers still remain underexplored compared to DNA methylation.

To address this gap, we performed a scoping literature review with the aim of delving deeper into the relationship of epigenomics and cervical cancer. This review aims to report additional documented epigenetic biomarkers linked to the disease, shifting the focus from nuclear DNA methylation to other less explored epigenetic mechanisms. Despite viral (HPV) epigenomics being an important factor to disease onset and progression, this review focuses on host epigenetic mechanisms due to their particular clinical relevance. A clinical biomarker based solely on host epigenomics does not depend on HPV methylation status or genotype, thereby broadening its applicability across diverse patient populations. The most prominent example of such biomarkers is the GynTect^®^ methylation marker panel which is based solely on the host genes FAM19A4 and miR124-2 [16]. Highlighting these alternative epigenetic mechanisms, our study aims to broaden the field of research for new biomarkers that can help improve early detection and personalized disease management strategies for cervical cancer.

## 2. Materials and Methods

To ensure a comprehensive search strategy for our study, we performed four independent literature searches, one for each of the following epigenetic modifications related to cancer: (1) mtDNA, (2) histone modifications, (3) non-coding RNA, and (4) repetitive elements methylation. All four literature searches were performed using the Google Scholar database in February 2025, following the guidelines outlined in the PRISMA (Preferred Reporting Items for Systematic Reviews and Meta-Analyses) extension for scoping reviews [17] and included all publications that were published until 31 January 2025. The exact search queries used in our methodology can be seen in Table 1 and the PRISMA extension checklist for scoping reviews is available in Appendix A. The results for each search were collated as a list in a separate Excel file (Appendix A) using Harzing’s Publish or Perish software version 8.9.4554.8721 [18]. The choice of keywords for each search was determined based on the corresponding epigenetic alteration and the technology required to measure it, e.g., “EPIC” for DNA methylation and “CHIP” for histone modifications. Each search included the term “cervical cancer” in order to limit the results to the particular disease. The choice of the Google Scholar database ensured that all the keywords would be searched inside the title, abstract, and main text of each manuscript in contrast to most databases where the search is limited to title and abstract. The data extracted from the Publish or Perish software included the following: No of citations, Authors, Title, Year, Source, Publisher, Article URL, Google Scholar citation URL, Type of document, Full text URL, and DOI. When full text URL or DOI was not available, a manual search for the manuscript was conducted.

Each list was used separately for the study selection process, which had the following steps: (1) duplicate removal, (2) screening of titles and abstracts, and (3) full text assessment. The final study selection was based on predefined inclusion and exclusion criteria.

Inclusion criteria for each selection were (1) being a primary research paper (not a review, book, thesis, or conference proceeding) producing its own data, (2) published in English, and (3) reported an association of cervical cancer with the corresponding epigenetic modification (Table 2). Every study selection process was performed independently by one reviewer and verified by two others.

In order to assess the number of publications per year and provide a bibliometric figure, all lists were imported into an R environment (version 4.4.1) and processed with a custom script (Appendix A).

## 3. Results

### 3.1. Study Selection Process

For mtDNA’s role in cervical cancer, several different combinations of search terms were examined. The final search query that returned the most relevant results was “HPV” “methylation” AND (“cervical cancer” OR “cervix”) AND (“mtDNA” or “mitochondrial DNA”) AND (“EPIC” OR “450K” OR “450 platform” OR “27K” OR “27 platform” OR “bisulfite sequencing” OR “pyrosequencing”). This final search query returned a total of 62 matches. Sources were discarded if they did not show insights into cervical cancer or did not study mitochondrial but only nuclear DNA methylation and/or if they did not investigate methylation in relation to cervical cancer cells. The number of publications that were discarded from the initial pool of matches can be seen in Figure 1. After filtering, only two relevant publications that described the impact of mitochondrial methylation in cervical cancer [19,20] were used in our review.

For histone modifications the search query with the most relevant results was [intitle:cervical cancer] “HPV” “histone modification” “epigenetic” AND (“cervical cancer” OR “cervix”) AND (“CHIP” OR “Chromatin Immunoprecipitation” OR “Mass spectrometry”), returning 54 matches. Sources were discarded if they did not show insight into histone modification in cervical cancer or if they were not primary research papers. From the remaining matches, 12 publications were relevant to histone modification in cervical cancer. The histone protein modifications identified in these studies were methylation, demethylation, acetylation, and deacetylation. The workflow of this query can be seen in Figure 2.

For non-coding RNAs the search query with the most relevant results was [intitle:cervical cancer] [intitle:RNA] “HPV” “RNA” “biomarker” “epigenetic” AND (“cervical cancer” OR “cervix”). The final search term returned 65 matches. Sources were discarded if they did not show insight into RNA mediated epigenetics in cervical cancer or if they were not primary research papers. After filtering, 30 publications were found to be relevant and included in our review. From these 30 publications, 21 were on lncRNA, 5 were on circRNA, 1 was on piRNA, and 3 were on miRNA. The workflow of this query can be seen in Figure 3.

For DNA methylation in repetitive elements the search query with the most relevant results was [intitle: “cervical cancer”] “methylation” AND (“Transposable element” OR “Transposons” OR “Retrotransposons” OR “Alu” OR “LINE-1” OR “SINE” OR “LTR” OR “Non-LTR” OR “repetitive elements”). The final search query resulted in 29 matches. However, 25 were discarded as they did not show insight into transposable elements in cervical cancer or were not primary research papers. The final number of manuscripts used in our study was four, as seen in Figure 4.

Notably, while the results from our search queries included articles published from 2000 onwards, the articles that were selected for our study range from 2010 to 2025 (Table 3). As seen in Figure 5 a big surge of publications on these epigenetic modifications occurs after 2015.

### 3.2. Methylation in Mitochondrial DNA

MtDNA methylation influences the expression of mtDNA genes, which are part of pathways in biogenesis, the electron transport chain, and oxidative phosphorylation [21,22]. Modified methylation patterns, such as low level of total nDNA methylation, hypermethylation of promoters of tumor suppressor genes, and hypomethylation of onco- gene promoters, are often observed in cancer [22]. Cancer cells often display a lower mtDNA copy number, which results in them being identified as differentiated from the cellular system. Conversely, global mtDNA demethylation has been shown to induce cellular differentiation by facilitating the expansion of mtDNA copy numbers and inhibiting tumorigenesis. While the literature on the role of mitochondrial DNA methylation on cervical cancer progression is sparse, disruptions in mtDNA methylation patterns are believed to be relevant in cervical cancer initiation and progression, since mitochondrial dysfunction contributes to carcinogenesis [22].

Sun et al. investigated methylation patterns of mtDNA amongst others [19]. They observed hypomethylation in mtDNA in cervical samples, but could not rule out sampling bias, a lack of methylase activity in mitochondria, or lack of association with cervix cancer, and thus argue that further studies are needed. Another study by Menga et al. discovered that the overexpression of the SLC25A26 gene, which encodes the mitochondrial carrier that catalyzes the import of S-adenosylmethionine (SAM) into the mitochondrial matrix and is required for mitochondrial methylation processes, is downregulated in cervical cancer cells [20]. This downregulation was linked to gene promoter hypermethylation.

Based on the observation of reduced SLC25A26 expression in cervical cancer cells, they established that mtDNA and, in particular, the D-loop control region is hypermethylated due to higher SAM mitochondrial levels.

### 3.3. Histone Modifications

In eukaryotes, genomic DNA is packaged into chromatin. Chromatin is made from nucleosomes, each one of which consists of around 150 base pairs of DNA wrapped around a histone octamer. A histone octamer consists of two copies of each of the four core histone proteins which are H2A, H2B, H3, and H4 [23]. Linker DNA connects nucleosomes, and a linker histone H1 may be bound in this region [24]. Histone proteins can go through post-translational modifications that can affect gene expression. These proteins can go through a variety of modifications such as methylation, acetylation, and phosphorylation, which can occur on many different amino acid residues [25]. Aberrant histone modification patterns have been identified in cancer [26], which makes them potential candidates for use as biomarkers for cervical cancer

Zhang et al. [7] identified that HPV18 can cause Enhancer of zeste homolog 2 (EZH2), a histone methyltransferase, to be overexpressed. This in turn led to the increased expression of H3K27me3, the tri-methylated histone H3 at lysin 27. This histone modification causes DNA (cytosine-5) methyltransferase 3A (DNMT3A) to be downregulated, which ultimately causes the overexpression of Tim-3/galectin-9. Tim-3 and galectin-9 are costimulatory factors that negatively regulate Type 1 T helper (Th1) immunity. Zhang et al. conducted a further study [27] and identified that the histone methyltransferase, SUV39H1, upregulates the expression of H3K9me3, the tri-methylated histone H3 at lysin 9, at the DNMT3A promoter region. This histone modification had the opposite effect, causing the overexpression of DNMT3A in cervical cancer. A study by Chen et al. [28] reported that selenium treatment for cervical cancer inhibits the expression of histone demethylases JMJD3 and UTX, leading to an increase in H3K27me3 expression.

A study by Ou et al. [29] reported that HPV16 E7 causes lysine-specific demethylase 2A (KDM2A) to be upregulated in cervical cancer. This causes the demethylation of histone H3 lysine 36 (H3K36), leading to the repression of the miR-132 microRNA. However, miR-132 typically plays a role in repressing cervical cancer invasion and proliferation. This shows that HPV16 E7-induced demethylation of H3K36 may play a role in promoting cervical cancer progression. A further study [30] was conducted which reported that HPV16 E7 also upregulates lysine-specific demethylase 5A (KDM5A) expression. This leads to the repression of microRNA miR-424-5p via demethylation of H3K4me2/3 and ultimately promotes cervical cancer progression. Liu et al. [31] found that lysine-specific demethylase 1 (LSD1), a histone demethylase, is recruited to the Vimentin promoter and demethylates H3K4me1 and H3K4me2. The study also reported that HPV16 E7 enhances LSD1 expression. Ectopic expression of LSD1 promotes cervical cancer cell invasion and metastasis [31].

Beyer at al. [32] analyzed the expression of acetylated histone H3 at lysine 9 (H3K9ac) and of tri-methylated histone H3 in lysine 4 (H3K4me3) in order to uncover their prognostic relevance in cervical cancer. These specific histone modifications were focused on due to their high metastatic potential. It was identified that H3K9ac expression was correlated with low grading in cervical cancer patients. H3K4me3 expression in cervical cancer was correlated with advanced T-status, a large tumor size, and poor prognosis [32]. This study identified histone acetylation and histone methylation within cervical cancer, further supporting the potential of histone modification as disease biomarkers. Pan et al. [33] found that activating AMP-activated protein kinase (AMPK) leads to the hyperacetylation of histone H3 at lysine 9 (H3K9) via the PCAF acetyltransferase. This modification leads to the transcriptional activation of tumor suppressor genes [33].

Feng at al. [34] investigated the correlation between histone acetylation and tumor suppressor gene (TSG) expression in cervical cancer. Retinoic acid receptor B2 (RARb2) and E-cadherin are both encoded by their respective TSGs. The study showed that the absence of Histone 3 acetylation (AcH3) was directly associated with poor histological differentiation and nodal metastasis. Furthermore a reduced expression of RARb2 and E-cadherin was observed in clinical tumor samples. Ultimately it was identified that histone deacetylation is associated with TSG silencing and cervical cancer progression. The use of targeted therapy with histone deacetylase (HDAC) inhibitors was suggested in order to restore TSG expression [34].

Du et al. [35] revealed that the transcription factor ETS1 recruits the histone acetyltransferase P300 and the histone methyltransferase WDR5 to the promoter region of the METTL3 gene. This process leads to increased acetylation of histone H3 at lysine 27 (H3K27ac) and tri-methylation of histone H3 at lysine 4 (H3K4me3), which leads to the activation of the METTL3 gene. Increased levels of this gene promote the progression of cervical cancer by enhancing m6A modification of TXNDC5 mRNA, which suppresses endoplasmic reticulum stress [35]. Another study [36] found that the transcriptional coactivator EP300 enhances H3K27ac at the promoter region of the NDUFA8 gene, which leads to its increased expression. Elevated levels of this gene promote mitochondrial respiration, which inhibits apoptosis in cervical cancer cells [36].

Yang et al. [37] identified that the histone acetyltransferase CSRP2BP is significantly overexpressed in cervical cancer tissues and associated with poor prognosis. CSRP2BP mediates the acetylation of histone H4 at lysine residues 5 and 12. This enhances the transcription of N-cadherin, and the upregulation of N-cadherin promotes epithelial–mesenchymal transition and metastasis of cervical cancer cells [37].

### 3.4. Non-Coding RNAs

Non-coding RNAs are RNAs that are not translated into proteins and instead have a housekeeping or regulatory role [38]. Recently, many studies have shown the significant role that non-coding RNAs have in epigenetics, where they can regulate gene expression [38]. There are several types of non-coding RNAs such as long non-coding RNA (lncRNA), circular RNA (circRNA), piwi-interacting RNA (piRNA), and microRNA (miRNA) [39].

#### 3.4.1. lncRNA

Several studies identified the overexpression of different long non-coding RNAs (lncRNAs) in cervical cancer. Yang et al. [40] found that PVT1, a lncRNA, was significantly increased in cervical cancer patients and was correlated with tumor size and lymph node metastasis or cervical cancer [40]. In another study PVT1 overexpression was found to inhibit microRNA-16 (miR-16) expression, which then promotes the cell cycle and inhibits cellular apoptosis, ultimately promoting cervical cancer development [41]. A study identified that the lncRNA FAM83H-AS1 was upregulated in early stages of cervical carcinogenesis, which could be a potential biomarker in order to identify the carcinoma early on [42]. Hu et al. [43] found that a hypoxia-induced lncRNA MIR210HG was excessively expressed in cervical cancer tissues [43]. Another study also identified the upregulation of a lncRNA. CCHE1 was found to be significantly correlated with large tumor size and poor survival [44]. Zhong et al. [45] found that the lncRNA AK001903 is upregulated in cervical cancer cells and tissues [45]. A study by Zhang et al. identified that the lncRNA CRNDE is overexpressed in cervical cancer tissues and several cervical cancer cell lines and was found to be positively correlated with poor overall survival in cervical cancer [46]. Another study by Zhang et al. found that the lncRNA KCNMB2-AS1 was significantly overexpressed in cervical cancer. They also identified N6-Methyladenosine (m6A), a post-transcriptional modification, in the lncRNA, ultimately enlarging its tumorigenic effect [47]. Chen et al. [48] found that the lncRNA Loc554202 was highly expressed in cervical cancer tissues and predictive of poor prognosis in patients [48]. Similarly, a study observed significantly high levels of the lncRNA PTTG3P in cervical cancer tissue and was associated with poor survival [49]. Xu et al. identified the lncRNA RP11-552M11.4 to be highly expressed in cervical cancer tumor tissues and observed patients with a high level of this lncRNA to have poor clinical outcomes [50]. Poor survival was also reported by a study identifying the lncRNA TP73-AS1 being upregulated in cervical cancer tissues [51]. Another study by Song et al. supports this, in which they showed the upregulation of TP73-AS1 both in cervical cancer tissues and cell lines [52]. The expression levels of another lncRNA named DLEU1 were reported to be significantly upregulated within serum-derived exosomes of cervical cancer patients compared to patients with cervical intraepithelial neoplasia and also healthy controls [53]. The same study also found that DLEU1 relative expression was significantly correlated with tumor size, deeper cervical invasion, higher pathological grade, and the presence of lymph node metastasis, suggesting its potential biomarker use in cervical cancer progression. Another study showed that lncRNA ABHD11 antisense RNA 1 (ABHD11-AS1) expression was abnormally high in cervical cancer cell lines HCC94, HeLa, C-33A, and CaSki compared to End1/E6E7 cells [54].

Despite many studies identifying an overexpression of several different lncRNAs, there are also lncRNAs whose downregulation is linked to cervical cancer. Zhang et al. identified the downregulation of lncRNA MEG3 and its association with poor prognosis in cervical cancer [55]. Another study supports this finding, reporting that MEG3 expression was downregulated in cervical intraepithelial neoplasia and squamous cell carcinoma tissues [56]. The lncRNA BDNF-AS was shown to be under-expressed in both cervical cancer cell lines and human cervical tumors [57]. A study identified that the expression of lncRNA LINC00899 was significantly downregulated in cervical cancer [58]. The lncRNA NKILA expression was found to be decreased in cervical cancer tissues and cell lines. It was observed that NKILA is involved with inhibiting the migration and invasion of cervical cancer cells [59]. Another study found that in a HPV-positive cervical cancer cell line (SiHa), there is a significant increase in DNA methylation at the promoter region of lncRNA MAGI2-AS3 compared to the HPV-negative cell line (C33A). This hypermethylation reduces the expression of lncRNA MAGI2-AS3 [60].

#### 3.4.2. circRNA

A few studies identified the aberrant expression of circular RNA (circRNA). The circRNA circYPEL2 was found to be significantly overexpressed in cervical cancer tissue [61]. A study identified the upregulation of circUBAP2 in cervical cancer tissues and cell lines, and its high expression predicted poor outcome [62]. The circRNA circRNA_101996 was overexpressed in cervical cancer patients and it was suggested that this upregulation assists cell proliferation and migration in cervical cancer [63]. A study identified the overexpression of circ_0003221 in cervical cancer tissues and cells [64]. There was also a study identifying the downregulation of a circRNA. Circ-ITCH was underexpressed in cervical cancer tissues and cell lines [65]. By overexpressing this circRNA, there was a significant inhibition of the tumorigenesis of cervical cancer, suggesting its potential as a therapeutic target [65].

#### 3.4.3. piRNA

Xie et al. identified that the piwi-interacting RNA (piRNA) piRNA-14633 was significantly upregulated in cervical cancer tissue and cells. The study demonstrated that the piRNA increases the expression of m6A RNA methylation, which contributes to the proliferation, migration, and invasion of cervical cancer [66].

#### 3.4.4. miRNA

A few studies identified the aberrant expression of miRNAs in cervical cancer. A study found that microRNA-145-3p (miR-145-3p) was downregulated in cervical cancer tissues compared with adjacent normal tissues [67]. Another study found that serum miR-199a was downregulated in high-risk HPV-positive cervical cancer patients compared to high-risk HPV-negative cervical cancer patients, identifying a relationship between miR-199a and high-risk HPV infection in cervical cancer patients. This study also found that miR-199a was inversely correlated with circMTO1. This circular RNA was upregulated in the serum of cervical cancer patients compared to healthy controls [68]. A study found that miR-205 levels were upregulated in cervical cancer tissues and cell lines compared to adjacent normal tissues. miR-205 was identified to be a target gene of lncRNA WT1-AS. In this study WT1-AS was found to be downregulated in cervical cancer tissues and cell lines [69].

### 3.5. Methylation in Repetitive Elements

Repetitive elements are estimated to comprise around 66% of the human genome and have been implicated in human disease due to their ability to cause instability by inserting copies in new genomic locations [70]. The insertion of these elements into specific genes has been observed in several types of cancer. One study has observed more Long Interspersed Element 1 (LINE-1) insertions within tumors than normal tissue [71]. The methylation of these elements, specifically Alu, is also believed to play a role in cancer progression, specifically in the Short-Interspersed element (SINEs) family. Alu hypomethylation was observed within the promoter region of the *MIEN1* gene (Migration and invasion enhancer 1), leads to gene expression, plays a role in cell migration and invasion, and is believed to play a role in the progression of PCa [72]. These elements play a role in cancer prognosis and progression and have potential use as biomarkers.

Curty et al. identified a relationship between the activity of DNA-methyltransferase 1 (DNMT1), which is involved in DNA methylation, and the expression of LINE-1 elements. The study found that lower levels of DNMT1 are associated with higher expression of LINE-1 retroelements in cervical cancer [73]. The same study examined retroelement activity in cervical cancer under various conditions: cancer types (squamous cell carcinoma and adenocarcinoma), human papillomavirus (HPV) strains (HPV18 and HPV16), and HPV infection scenarios (single vs. multiple infections). Among 103 retroelements analyzed, over half exhibited varying activity between squamous cell carcinoma and adenocarcinoma. Distinctions were observed in retroelement types (L1 and HERV) and families (e.g., HERV-K and HERV-H). Notably, L1 retroelements displayed heightened activity in cases of multiple HPV infections. Specific retroelements were exclusive to certain conditions, such as HPV infections, cancer types, or HPV strains [73]. The inverse correlation between DNMT1 levels and increased LINE-1 retroelement expression in cervical cancer, along with insights into diverse retroelement activities under various conditions, underscores the potential for utilizing transposable elements as prognostic markers and therapeutic targets in the management of cervical cancer.

Sen et al. found a connection between cervical cancer and the epigenetic regulation of transposable elements. Their findings indicate that global DNA hypomethylation, particularly within repetitive Alu sequences, is a characteristic of cervical cancer, especially in cases with episomal HPV16. This suggests that altered methylation patterns in transposable elements are implicated in their development and progression [74]. Additionally, the study found that cases with episomal HPV16 had significantly higher global host DNA hypomethylation than those with viral integration.

McCabe et al. looked at the STK11/LKB1 gene encoding a serine/threonine kinase that has been observed to be mutated in different cancers, including non-small cell lung, pancreatic, and melanomas [75]. It plays a crucial role in various cellular functions such as cell growth, cell cycle progression, metabolism, cell polarity, and migration. They identified homozygous deletions spanning 25–85 kb in the HeLa and SiHa cervical cancer cell lines. In HeLa cells, the deletion results from Alu-recombination-mediated deletion (ARMD) [75]. This suggests that the loss of genetic material, in this case, results from recombination events involving Alu elements. The specific Alu elements likely played a role in mediating the deletion of a segment of DNA, including the LKB1 gene, ultimately leading to the observed genetic alteration.

Nguyen et al. suggested that HPV integration into the genome can lead to the transcription of usually silent regions, particularly LINEs, SINEs, and LTRs, and this dysregulation may contribute to genomic instability and pathway disruption, potentially promoting tumorigenesis in HPV-associated cervical cancers [76].

## 4. Discussion

This scoping review aimed to provide a comprehensive overview of alternative epigenetic biomarkers associated with cervical cancer, shifting the focus from nuclear DNA methylation to other less-explored epigenetic mechanisms. However, as we recognize the importance of nuclear DNA methylation, we also included an in-depth exploration of specific genomic targets whose aberrant methylation is known to be associated with cancer: repetitive elements [77]. Our findings suggest that histone modifications, methylation of mtDNA, non-coding RNAs, and repetitive elements are all contributing factors to the altered epigenetic profile of cervical cancer and have the potential for biomarker development and cancer treatment.

### 4.1. Mitochondrial DNA Methylation: A Potential but Untapped Biomarker

While the role of methylation of nuclear DNA in cancer is well established [78], methylation of mtDNA is a relatively less explored area with sometimes conflicting results. Our literature review revealed scarce but interesting evidence that mtDNA methylation changes were associated with cervical cancer. Significantly, the D-loop control region was hypermethylated in cancer cervical cells due to reduced expression of the SLC25A26 gene, essential for mitochondrial methylation processes. In spite of one study questioning the generalizability of mtDNA methylation as a cancer biomarker, evidence for cervical cancer association suggests otherwise. The conflicting evidence puts emphasis on more focused research on mtDNA methylation and its functional role in CIN onset and cervical cancer progression.

### 4.2. Histone Modifications: A Reversible Contributor to Cervical Oncogenesis

Histone modifications are one of the most extensively studied epigenetic mechanisms associated with cancer. In our review multiple histone methylation and acetylation marks, such as H3K27me3, H3K9me3, H3K4me3, and H3K9ac, were found to be linked with cervical cancer, as they regulate critical cellular processes leading to tumorigenesis [79]. Such processes include, but are not limited to, tumor suppressor gene silencing, epithelial–mesenchymal transition, and T-helper immunity and mitochondrial function. Notably, it was demonstrated that HPV oncoproteins (notably E7) upregulate the expression of histone demethylases LSD1, KDM2A, and KDM5A, thus resulting in a viral-induced epigenetic disruption in the host cells. Such changes enhance cancer cell invasion, metastasis, and immune evasion.

However, histone modifications’ mechanisms are reversible, which grants them potential for therapeutic applications. The proof of concept for such applications has already been demonstrated by the anti-cancer effects of selenium and AMPK-activating treatments, as well as from the proposed application of HDAC inhibitors for reactivation of tumor suppressor gene expression.

### 4.3. Non-Coding RNAs: Expression as a Diagnostic and Prognostic Biomarker

Non-coding RNAs have proven to have potential as biomarkers of early diagnosis, prognosis, and progression of cervical cancer, depending on their expression levels. The upregulation of specific lncRNAs, including PVT1, FAM83H-AS1, MIR210HG, CCHE1, AK001903, CRNDE, DLEU1, and KCNMB2-AS1 in either tissues or cell lines, was observed in multiple studies, and associated with poor prognosis, large tumor size, and/or metastasis. On the other hand, lncRNAs such as MEG3, BDNF-AS, LINC00899, MAGI2-AS3, and NKILA were found to be downregulated in similar cases and resulted in a decrease in tumor-suppression activity. Interestingly, lncRNAs like MAGI2-AS3 were epigenetically repressed by promoter hypermethylation, which shows the connection between DNA methylation and non-coding RNA regulation. Similarly, circRNAs like circYPEL2, circUBAP2, and circRNA_101996 were overexpressed and were associated with tumor development, while circ-ITCH showed tumor-suppressing potential. The differential expression of miRNAs such as miR-145-3p and miR-205 has also been linked to cervical cancer status. Additionally miR-199a downregulation was linked to high-risk HPV infection. Some miRNAs have been observed to be differentially expressed in cervical cancer depending on other histone modifications. An example of this is the repression of miR-424-5p via the demethylation of H3K4me2/3. Finally, the recent findings of piRNA-14633 involvement in m6A methylation and cervical cancer progression reveal another epigenetic regulation mechanism that will need to be investigated in detail.

### 4.4. Hypomethylation of Repetitive Elements: Biomarker of Genomic Instability

The methylation profiles of repetitive elements such as LINE-1, Alu, and HERV have been associated with cervical cancer in multiple studies. The outcome of these studies is that global hypomethylation, especially in LINE-1 and Alu elements, is often related to cervical cancer, driven by HPV infection status and cancer type. This hypomethylation is causally connected with increased retroelement expression, which can lead to genomic instability and tumorigenesis. Additionally, the loss of DNMT1 function, a key component to the cell’s DNA methylation mechanisms, was shown to enhance such retroelement overexpression, which makes it a potential biomarker for cervical cancer progression. Similarly, Alu recombination events can lead to homozygous deletion of tumor suppressor genes, such as LKB1, in cervical cancer cell lines. These findings suggest that epigenetic modifications in repetitive elements have significant potential as a biomarker as they play an etiological role in tumor suppressor gene stability.

## 5. Conclusions

Our findings emphasize the complex involvement of epigenetic modifications in cervical carcinogenesis. Even though most of these biomarkers are still in the preclinical stage, they have enormous potential to be translated to the clinic. Furthermore, applying non-invasive sampling approaches (e.g., serum-derived exosomes for lncRNA detection) adds significantly to their potential as diagnostic reagents.

However, several limitations were noticed in the literature. Most of the studies that were included in this review had relatively small sample sizes, lacked longitudinal data, and/or were not validated in independent cohorts. As this is a scoping review, the main focus of our study was to report all epigenetic alterations, other than the nuclear DNA methylation, linked to cervical cancer. An additional meta-analysis is needed to evaluate the successive appearance of these alterations, including nuclear DNA methylation, and biological mechanisms, also in relation with HPV infection, by which these epigenetic alterations promote CIN lesions and cervical cancer.

Despite these limitations, the multitude of evidence accumulated by these studies makes it abundantly clear that these epigenetic mechanisms are considerable contributors to disease progression and, going forward, they should be included in future study designs. The two most important technological advancements that can further facilitate the study of these mechanisms are the advent of machine learning (ML) methodologies, and the progress of high-throughput sequencing techniques. There is currently a surge of machine learning methodologies becoming gradually more elaborate, such as neural networks (e.g., deep learning) that can capture complex relationships between model features (i.e., biomarkers), as well as explainable artificial intelligence (xAI) which can help interpret these intricate relationships. On the other hand, sequencing companies have been constantly improving their output yields and reducing their costs, making it even easier to investigate a large enough number of samples to allow for the training of robust ML models. In addition, the higher coverages achieved for a fraction of the cost can lead to accurately capturing methylation profiles of less abundant molecules such as mtDNA.

With these advancements in mind, any future works on these epigenetic biomarkers should be performed on large-scale, multiethnic cohorts to validate these biomarkers in various populations, while including a thorough functional analysis to identify the mechanistic relationships between epigenetic alterations and cervical carcinogenesis. This could potentially lead to the development of multi-omic marker panels involving DNA methylation (both nuclear and mitochondrial), histone modification, and non-coding RNA profiles for the early diagnosis and prognosis of cervical cancer.

## Figures and Tables

**Figure 1 ijms-26-09423-f001:**
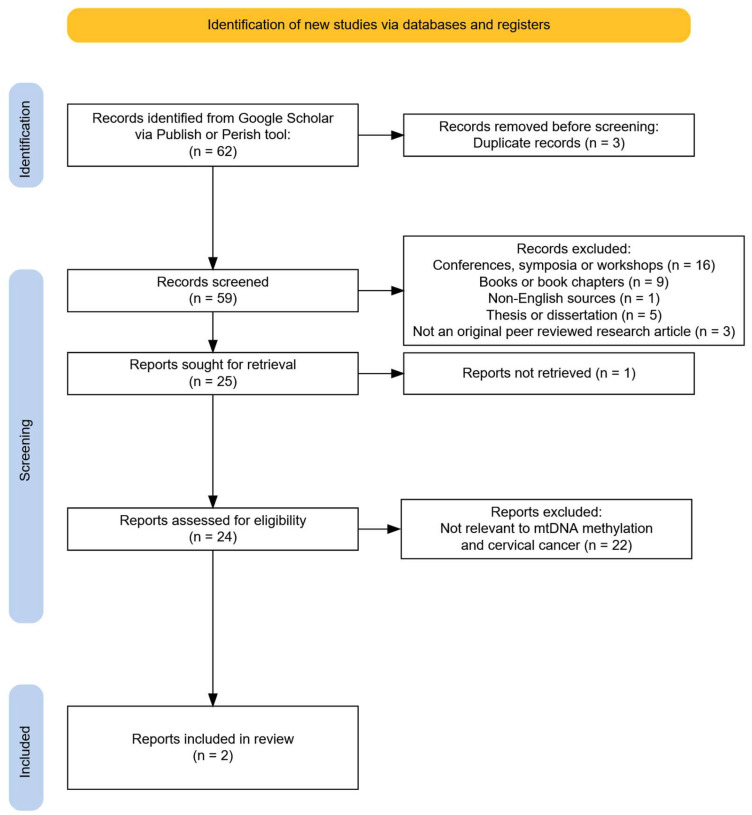
PRISMA flowchart on the mtDNA study selection process.

**Figure 2 ijms-26-09423-f002:**
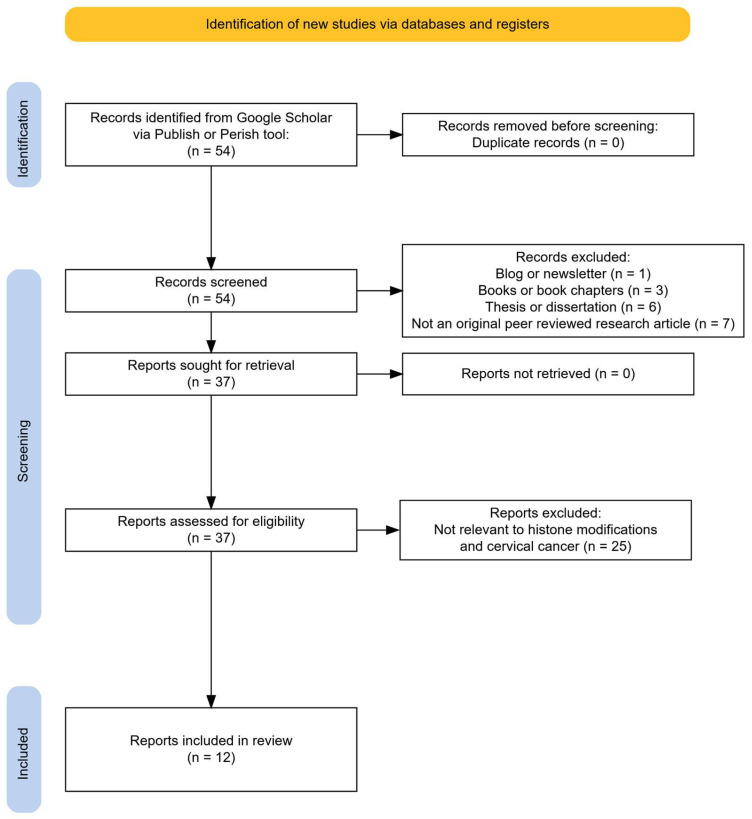
PRISMA flowchart on the histone modifications study selection process.

**Figure 3 ijms-26-09423-f003:**
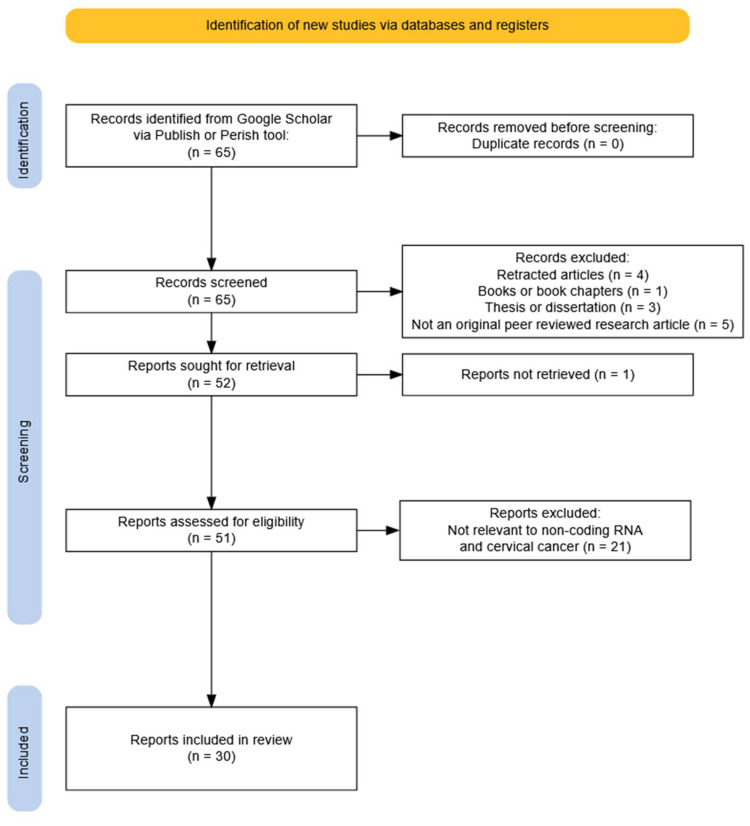
PRISMA flowchart on the non-coding RNA study selection process.

**Figure 4 ijms-26-09423-f004:**
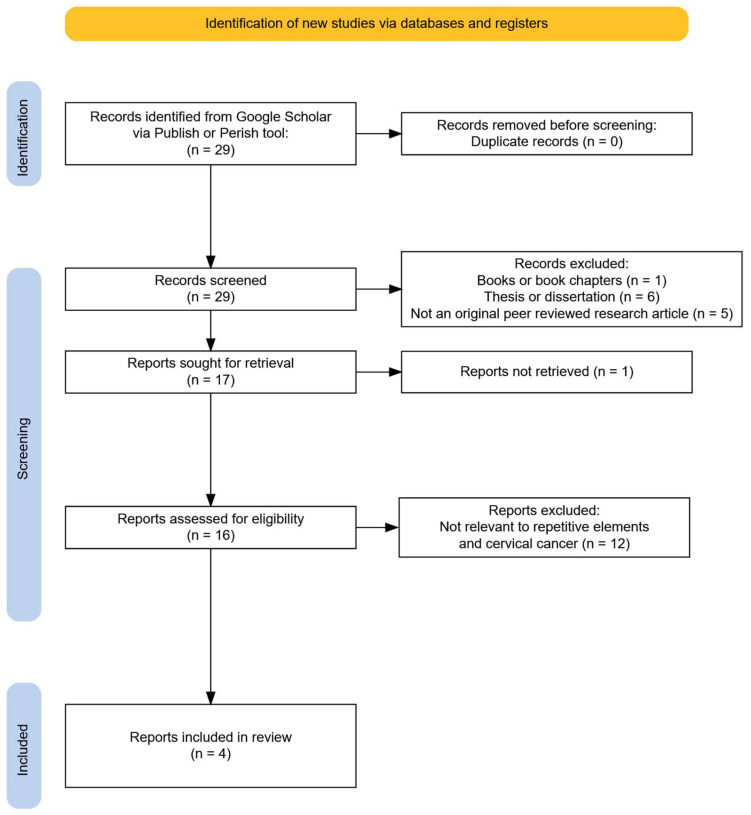
PRISMA flowchart on the repetitive elements study selection process.

**Figure 5 ijms-26-09423-f005:**
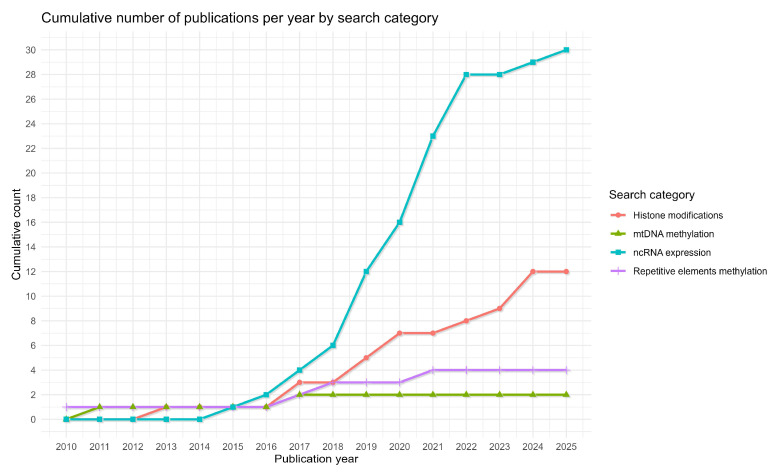
Publications per year for each epigenetic modification search.

**Table 1 ijms-26-09423-t001:** Search queries used in Google Scholar via the Publish or Perish tool.

Search Query in Google Scholar	Epigenetic Modification
“HPV” “methylation” AND (“cervical cancer” OR “cervix”) AND (“mtDNA” or “mitochondrial DNA”) AND (“EPIC” OR “450K” OR “450 platform” OR “27K” OR “27 platform” OR “bisulfite sequencing” OR “pyrosequencing”).	Mitochondrial DNA methylation
[intitle:cervical cancer] “HPV” “histone modification” “epigenetic” AND (“cervical cancer” OR “cervix”) AND (“CHIP” OR “Chromatin Immunoprecipitation” OR “Mass spectrometry”)	Histone modifications
[intitle:cervical cancer] [intitle:RNA] “HPV” “RNA” “biomarker” “epigenetic” AND (“cervical cancer” OR “cervix”)	Non-coding RNA expression
[intitle: “cervical cancer”] “methylation” AND (“Transposable element” OR “Transposons” OR “Retrotransposons” OR “Alu” OR “LINE-1” OR “SINE” OR “LTR” OR “Non-LTR” OR repetitive elements”)	Repetitive elements methylation

**Table 2 ijms-26-09423-t002:** Inclusion and exclusion criteria.

Inclusion Criteria	Exclusion Criteria
Primary research paper	Review
Manuscript published in English	Book or book chapter
Study reported association between cervical cancer and epigenetic modification	Thesis or dissertation
	Proceedings from conferences, symposia or workshops
	Article in blog or newsletter
	Retracted study

**Table 3 ijms-26-09423-t003:** Number of relevant manuscripts and period of publication from each search query.

Epigenetic Modification Related to Cervical Cancer	Number of Manuscripts	Publication Period
Mitochondrial DNA methylation	2	[2011–2017]
Histone modifications	12	[2013–2024]
Non-coding RNA expression	31	[2015–2025]
Repetitive elements methylation	4	[2010–2021]

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
