# Peer review of "Epigenetic Biomarkers for Cervical Cancer Progression: A Scoping Review"

_ijms, 2025, doi:10.3390/ijms26199423_

Round 1
Reviewer 1 Report
Comments and Suggestions for Authors
I consider that some sections are not entirely complete, for example non-coding RNA (miRNA and snoRNA could be incluided). Also a section of HPV genome methylation as progression biomarker for cervical cancer should be added.
Author Response
Comment: I consider that some sections are not entirely complete, for example non-coding RNA (miRNA and snoRNA could be incluided). Also a section of HPV genome methylation as progression biomarker for cervical cancer should be added.
Reply: Thank you for noticing this. After re-examining the non-coding RNA studies we noticed that 3 of them were erroneously classified as “non relevant” while they indeed reported miRNA as a potential biomarker for cervical cancer. We have now added an miRNA section (lines 383-395). Figure 3 and Supplementary Table 3 are also updated to reflect the new number of studies included in our manuscript. Regarding snoRNA, we did not find any studies that would report its role as a potential epigenetic biomarker for cervical cancer. Regarding HPV genome methylation, despite being an important factor to disease onset and progression, this review focuses on host epigenetic mechanisms due to their particular clinical relevance. A clinical biomarker based solely on host epigenomics does not depend on HPV methylation status or genotype, thereby broadening its applicability across diverse patient populations. This is now explained in the Introduction section (lines 89-93).
Reviewer 2 Report
Comments and Suggestions for Authors
- 1.Language expression and logical coherence: Some sentences are too long, and it is recommended to split them to improve readability (such as the paragraph on histone modification in Section 3.3); Some concluding statements lack support, such as "mtDNA methylation is a potential biomarker" should be expressed more carefully, reflecting the limited nature of existing research.
2.Methodological details: It should be clarified whether only Google Scholar was used for the search or whether it was combined with databases such as PubMed and Scopus to avoid search bias. Additionally, it should be specified whether quality assessments (e.g., ROBIS, QUADAS, etc.) were conducted to enhance methodological transparency.
3.Structure of the results section: Section 3.2.1, which discusses the prevalence of mitochondrial DNA in cancer, appears somewhat redundant and could be streamlined to focus more directly on cervical cancer.
4.Incomplete abbreviation list: The text uses multiple abbreviations not listed in the abbreviation table (e.g., CUT&Tag, ATAC-seq, EP300, METTL3, etc.). It is recommended to add them in full.
Author Response
1.Language expression and logical coherence: Some sentences are too long, and it is recommended to split them to improve readability (such as the paragraph on histone modification in Section 3.3); Some concluding statements lack support, such as "mtDNA methylation is a potential biomarker" should be expressed more carefully, reflecting the limited nature of existing research.
Reply: We have reviewed the entire manuscript and improved the language expression. We gave extra focus on Section 3.3 to improve the readability. We have also expressed statements like "mtDNA methylation is a potential biomarker" more carefully.
2.Methodological details: It should be clarified whether only Google Scholar was used for the search or whether it was combined with databases such as PubMed and Scopus to avoid search bias. Additionally, it should be specified whether quality assessments (e.g., ROBIS, QUADAS, etc.) were conducted to enhance methodological transparency.
Reply: We have now clarified that only Google Scholar was used and explained that it was because it ensured that that all the keywords would be searched inside the title, abstract and main text of each manuscript in contrast to most databases (lines 116-118). Bias risk assessment tools were not used in this study as this is a scoping review and tools such as ROBIS and QUADAS are designed for systematic reviews.
3.Structure of the results section: Section 3.2.1, which discusses the prevalence of mitochondrial DNA in cancer, appears somewhat redundant and could be streamlined to focus more directly on cervical cancer.
Reply: Sections 3.2.1 and 3.2.2 were merged with the former being truncated as a small introduction to the latter, thus focusing more directly on cervical cancer (lines 209-220).
4.Incomplete abbreviation list: The text uses multiple abbreviations not listed in the abbreviation table (e.g., CUT&Tag, ATAC-seq, EP300, METTL3, etc.). It is recommended to add them in full.
Reply: The abbreviation table has been updated to include all abbreviations in the main text.
Reviewer 3 Report
Comments and Suggestions for Authors
Comments are proposed in the joined document

Author Response
Referee's comments:
The article of Ladoukakis and coauthors, untitled, "Epigenetic biomarkers for cervical cancer progression: A scoping review" reports a synthesis of the literature that concern epigenetic mechanisms related to cervical cancer other than nuclear DNA methylation largely published up today. The secondary objective of this review is to identify some new biomarkers related to cervical cancerous lesions (CIN2 and 3) at early stage. Indeed, HPV testing in primary screening has a very poor positive predictive value (less than 20%), its interest is its negative predictive value reaching nearly 100%. Pap smear as a triage test increase this PPV but still has a lack of sensitivity (about 60%), cervical CIN lesions thus may not be early detected. Therefore, we need some other biomarkers for early screening of CIN lesions.
The literature searches followed the PRISMA guidelines. The key words used for the search queries are reported in table 1, the choice of these words are not defined. Indeed, this choice is fundamental, some articles may not be find if some key words are missing. For example, only 2 studies upon 63 that were matches for the first item (mtDNA methylation) were identified. How were defined these key words?
Reply: The choice of keywords for each search was determined based on the corresponding epigenetic alteration and the technology required to measure it, e.g. “EPIC” for DNA methylation and “CHIP” for histone modifications. Each search included the term “cervical cancer” in order to limit the results to the particular disease. The choice of the Google Scholar database ensured that that all the keywords would be searched inside the title, abstract and main text of each manuscript in contrast to most databases where the search is limited to title and abstract. This is now mentioned in the manuscript (lines 112-118)
This work reports several epigenetic alterations related to cervical cancer as a list of altered genes, however little is reported with combination of altered genes. Furthermore, little associations to HPV oncogenes are reported. No kinetic studies are reported to evaluate the successive appearance of theses alterations. It is thus difficult to propose epigenetic mechanisms related to the appearance of CIN lesions.
The authors should propose as a conclusion a schematic synthesis of these mechanisms in relation with HPV infection and CIN lesions if possible.
Reply: As this is a scoping review, the main focus of our study is to report all epigenetic alterations, other than the nuclear DNA methylation, linked to cervical cancer. An additional meta-analysis is needed to evaluate the successive appearance of these alterations, in addition to nuclear DNA methylation and the mechanisms in relation with HPV infection and CIN lesions. This is now mentioned in the manuscript (lines 533-536)
Other comments:
Lane 51, HPV screening tests allows the monitoring but not the identification of CIN lesions, it is an indirect biomarker.
Reply: This is now amended to indicate that Pap smear allows the monitoring of CIN lesions while HPV screening tests is an indirect biomarker of disease risk. (lines 51-54)
Lane 59, the host immune response should be also considered as factors. Is there a relation with epigenetic alterations ?
Reply: The host immune response has been added to the Introduction as a factor affected by epigenetic alterations and an example has been given specific to cervical cancer. (lines 61-64)
Table 1 could be completed with the number of articles found and the period of publication. Indeed, it is reported by the authors that the database was closed in February 2025. In the references, the articles year of publication goes from 2005 to 2025, most of them were after 2015. The authors could give a bibliometric figure in the results.
Reply: We added an additional table in the Results section with the number of articles found and the period of publication (Table 3, lines 199-200). We have also provided an additional figure with the cumulative number of publications per year (Figure 5, lines 205-207). Finally, we note in the manuscript that while the results from our search queries included articles published from 2000 onwards, the articles that were selected for our study, range from 2010 to 2025 (lines 194-196).
Figure 1 reports 62 records and in lane 124, 63 matches, there is one missing? In this figure, 22 out 24 are not relevant to mtDNA methylation, what were these articles (wrong key words) ?
Reply: We have corrected the manuscript to note that there were 62 matches (new line 149). The key words were not the issue rather than the lack of research on mtDNA methylation and cervical cancer.
Paragraphs 3.2 to 3.5 are part of the discussion and not only results.
Reply: We deliberately describe in detail the findings of our searches for each one of the epigenetic modifications in paragraphs 3.2 to 3.5. We consider these to be part of our results as they are the direct outcome of our literature review. We summarise these findings in the Discussion in paragraphs 4.1 – 4.4. Our aim is to provide a summarised Discussion section that is easy for the reader to follow while providing details about every study we included in the Results section.
Discordance in studies: DNMT3A downregulated lane 217 and overexpressed in lane 224 ?
Reply: These two studies from Zhang et al. report two different histone modifications which have an opposite effect in DNMT3A. In the first study. the tri-methylated histone H3 at lysin 27 (H3K27me3) causes DNMT3A to be downregulated which ultimately causes the overexpression of Tim-3/galectin-9. In contrast, in their second study they report that the tri-methylated histone H3 at lysin 9 (H3K9me3), causes the overexpression of DNMT3A and consequently the underexpression of Tim-9/galectin-9. We now note in the manuscript that these two histone modifications have the opposite effect on DNMT3A (lines 252-253).
PVT1 overexpressed in lane 290 and reported in the downregulation lncRNA in lane 328?
Reply: This is now corrected and added in the upregulation part of the lncRNA section (line 311-316).